# Verification of Tensile Force Estimation Method for Temporary Steel Rods of FCM Bridges Based on Area of Magnetic Hysteresis Curve Using Embedded Elasto-Magnetic Sensor

**DOI:** 10.3390/s22031005

**Published:** 2022-01-27

**Authors:** Won-Kyu Kim, Junkyeong Kim, Jooyoung Park, Ju-Won Kim, Seunghee Park

**Affiliations:** 1Equipment Group, Industrial Materials Team 2, Materials Division, Samsung C&T Corporation, Seoul 05510, Korea; kwk1104@gmail.com; 2Department of Convergence Engineering for Future City, Sungkyunkwan University, Suwon 16419, Korea; 3Safety Inspection for Infrastructure Laboratory (SIIL), Advanced Institute of Convergence Technology, Suwon 16229, Korea; 4Department of the Civil, Architectural and Environmental System Engineering, Sungkyunkwan University, Suwon 16419, Korea; mitjy26@gmail.com; 5Department of Safety Engineering, Dongguk University-Gyeongju, Gyeongju 38066, Korea; juwon@dongguk.ac.kr; 6School of Civil, Architectural Engineering & Landscape Architecture, Sungkyunkwan University, Suwon 16419, Korea; 7Technical Research Center, Smart Inside AI Co., Ltd., Suwon 16419, Korea

**Keywords:** free cantilever method (FCM), temporary steel rod, tensile force, embedded Elasto-Magnetic (EM) sensor, magnetic hysteresis curve, temperature compensation

## Abstract

The free cantilever method (FCM) is a bridge construction method in which the left and right segments are joined in sequence from a pier without using a bottom strut. To support the imbalance of the left and right moments during construction, temporary steel rods, upon which tensile force is applied that cannot be managed after construction, are embedded in the pier. If there is an excessive loss of tensile force applied to the steel rods, the segments can collapse owing to the unbalanced moment, which may cause personal and property damage. Therefore, it is essential to monitor the tensile force in the temporary steel rods to prevent such accidents. In this study, a tensile force estimation method for the temporary steel rods of an FCM bridge using embedded Elasto-Magnetic (EM) sensors was proposed. After the tensile force was applied to the steel rods, the change in tensile force was monitored according to the changing area of a magnetic hysteresis curve, as measured by the embedded EM sensors. To verify the field applicability of the proposed method, the EM sensors were installed in an FCM bridge pier under construction. The three sensors were installed in conjunction with a sheath tube, and the magnetic hysteresis curve was measured over nine months. Temperature data from the measurement period were used to compensate for the error due to daily temperature fluctuations. The estimated tensile force was consistent with an error range of ±4% when compared with the reference value measured by the load cell. Based on the results of this experiment, the applicability of the proposed method was demonstrated.

## 1. Introduction

As the construction industry develops, the importance of effective and efficient maintenance techniques for the structures is increasingly emphasized. In the case of bridges, the need for more advanced maintenance technology is gradually increasing as construction methods progress [1,2]. Prestressed Concrete (PSC) has been used in numerous bridges since the late 1960s [3] when the post-tension method was first used in bridge construction. PSC is characterized by a higher tensile strength than plain concrete because the initial stress is pre-applied to the concrete. In addition, PSC offers the advantage that the construction can be performed at a low cost, which is increasingly demanded in the bridge construction industry [4].

The free cantilever method (FCM) is a construction method for PSC bridges that does not install scaffolding systems under the bridge and completes the superstructure of the bridge by sequentially joining the segments to form a span by post-tensioning and balancing them left and right from each pier using special erection equipment. The FCM has advantages in terms of construction conditions and its period. As it does not use a shore or a scaffold, the bridge can be constructed under extreme conditions, such as in deep valleys or at sea. Furthermore, the construction period can be shortened in the case of using the precast concrete method, where manufactured girders are assembled at the construction site, rather than the in situ concrete method. Thus, the FCM is consistently used for bridge construction. However, considering the characteristics of the FCM, which joins segments from side to side, an unbalanced moment may occur owing to the load generated during construction, as shown in Figure 1.

To prevent such a problem, temporary steel rods are embedded in the pier table. Then, calculated tensile forces are introduced to withstand the loads and their resulting moments during the construction stage. However, because the temporary steel rods are embedded in the pier table, it is almost impossible to visually observe and determine whether the tensile forces introduced to the rods are being maintained properly. If the signs of destruction or loss of tension are not recognized at the early stages, and consequential actions are not taken in a timely manner, accidents, such as sudden collapses and falls, may occur during construction. Therefore, a Non-Destructive Test (NDT) technique, which can be applied to the temporary steel rods during construction, is necessary to reduce the potential accidents.

Recently, a great deal of research has been conducted on measuring steel tension, such as the tensile force estimation method of a PS tendon using a Fiber Bragg Grating (FBG) sensor [5,6] or an Elasto-Magnetic (EM) sensor [7,8,9] and a tensile force measurement method of steel wire using a natural frequency measurement [10]. In addition, tensile force measurements of unbonded steel wires using longitudinal guided ultrasonic measurement techniques [11,12] and magnetic flux transmission monitoring techniques using magnetic circuits [13] have been studied. A method that uses an acoustoelastic theory was proposed to evaluate the prestress levels in post-tensioned steel strands employing changes in longitudinal stress wave velocity [14,15]. The aforementioned studies showed the estimated tensile force from the sensor response according to the introduced tensile force. However, the changes in the measured value due to environmental changes, such as temperature, were not taken into consideration with a sufficiently long observation period. A study that considered the temperature condition of the target structure was also proposed using various sensors. An SHM system for a long-span, cable-membrane structure was also proposed by Tang et al. [16]. It was applied to monitor the structural static responses, structural vibration, and environmental effects of the structure using various types of sensors, including a magnetic flux sensor, an FBG strain gauge, an FGB thermometer, an accelerometer, and so on. A study to estimate the girder deflection under thermal actions was conducted for a cable-stayed bridge [17]. Although those studies showed the correlation between the temperature distribution on the members of the bridge and the deflection of the girder by thermal actions, this study focuses on the change in the signal of the sensor, which measures the tension force as the change in temperature of the target member leads to the change in the obtained signals. Therefore, in this study, a technique was proposed for estimating the tensile force during the construction of an FCM bridge using embedded EM sensors to monitor the magnetic hysteresis of the temporary steel rods, considering the temperature effect on the sensors continually. The embedded EM sensor consists of two coils and a bobbin. The primary coil generates a magnetic field after magnetizing the steel rods, and the secondary coil serves to measure the magnetic flux generated in the magnetized steel rods. The magnetic hysteresis curve generated from the measured magnetic flux varies depending on the change in the magnetic field according to the change in the tension of the steel rods [18,19]. It leads to the change of the area of the hysteresis curve. The tensile force was estimated by calculating the changing area. However, the sensor coil, being made of copper, is very sensitive to the change of temperature [20] and, thus, affects the sensor response. The signals from the sensors may not be interpreted accurately without reflecting on this phenomenon. Therefore, in this study, a temperature compensation method was applied for the correction of the errors caused by the temperature change.

## 2. Theoretical Background and Methods

### 2.1. EM Sensor

The EM sensor used in this paper is composed of a part for inducing voltage to generate magnetic fields and a part for measuring it, as shown in Figure 2. The coils are wound on the outer surface of the bobbin, and insulation covers are wrapped between each coil. The coils pulled out of the bobbin are connected to the data-acquisition device through connectors. A protection cover is on the outermost surface of the EM sensor, and it protects the sensors from the concrete pouring. For the attachment of the EM sensor to the sheath around the temporary steel rod, conchoids are on the inner surface of both ends of the bobbin.

### 2.2. Prestress Loss in PSC Bridge

Prestress is introduced to the PSC bridge at each major construction stage of the bridge superstructure. The loss of prestress introduced to the structure has various causes, and the type of the loss is usually classified into immediate loss and long-term loss [21]. Immediate loss occurs when the prestress is introduced, and it includes the loss due to friction between the PS steel and the sheath pipe, the anchorage slip, and the elastic deformation of the concrete. Long-term loss refers to the loss that occurs over time after the introduction of the prestress, and it is related to the creep and drying shrinkage of the concrete and the relaxation of the PS steel.

### 2.3. Tensile Force Estimation through Measuring Area of Magnetic Hysteresis Curve

In this study, the magnetic hysteresis curves were measured using embedded EM sensors to monitor the change in the tensile force of the temporary steel rods during the construction of an FCM bridge. Three embedded EM sensors were installed with a sheath pipe outside the steel rod used for the construction. As shown in Figure 3, when a voltage is applied to the primary coil of the installed EM sensor, the magnetic field of the steel rod becomes saturated. On the other hand, when a reverse voltage is applied, the direction of magnetic field in the steel rod is reversed, creating a magnetic hysteresis curve (B–H Loop) [22]. However, the shape of the hysteresis curve begins to change when some amount of force is applied to induce stresses inside the steel rod [23]. The letter A in Figure 3 shows a state of magnetization generated in the absence of stress, while the letter B indicates a magnetization state after stress is applied [24].

The magnetic hysteresis curve measured by this principle represents the relationship between the strength of the magnetic field and the magnetic flux density of a ferromagnetic material, which is used to indicate the magnetic property of the ferromagnetic material. In addition, when the tensile force introduced to the steel rod changes, the magnetic properties change by the inverse magnetostriction effect, leading to the consequential change of the magnetic hysteresis curve. When the tensile force increases, the magnetic flux density increases, leading to an increase in the magnetic flux leakage. As the magnetic flux leakage increases, the area of the magnetic hysteresis curve increases. Therefore, it is possible to estimate the state of the introduced tensile force by measuring the area increase and decrease of the magnetic hysteresis curve for the temporary steel rods.

### 2.4. Temperature Compensation Method

A temperature compensation technique was adopted to correct the measurement errors due to the temperature changes during the measurement. Temperature compensation is essential because steel materials, including the temporary steel rods used in this study, are greatly affected by external forces and temperature changes [25,26,27]. Such compensation techniques are broadly divided into software-based methods and hardware design methods. However, the methods through hardware compensation have a limitation in their use in the field because they suffer from poor reliability due to their inability to cope with design errors that may occur during production [28]. Therefore, in this study, a software compensation technique using a quadratic polynomial was applied. The polynomial compensation technique is widely used as a temperature compensation technique for measurement data. The quadratic polynomial employed in this study is represented by the following regression equation [27]:(1)$$F_{p}=a_{00}+a_{10}V_{T}+a_{01}T+a_{20}V_{T}^{2}+a_{11}V_{T}\cdot T+a_{02}T^{2},$$
where *F_p_* is the tensile force predicted by the quadratic polynomial fitting; *V_T_* is the applied voltage; *T* is the on-site temperature; and *a*_00_, *a*_10_, *a*_01_, *a*_20_, *a*_11_, and *a*_02_ are the second-order fitting coefficients. The change in the area of the graph of the measured magnetic hysteresis curve, using embedded EM sensors, was approximated to the second order through regression analysis. The temperature dependence was compensated for by substituting the temperature data measured in the field into the equation with their respective measurement times.

## 3. Field Experimental Results and Discussion

### 3.1. Experimental Setup

To apply the embedded EM sensors to the site, a tensile force monitoring experiment was conducted by measuring the magnetic hysteresis curve at the construction site of a PSC bridge where the FCM was applied. The test bridge was a box-type PSC girder bridge with a span of 640 m and a width of 24.51 m at a construction site in Asan city, Chungcheongnam-do, Republic of Korea. The steel rods applied to the site were circular rods with a diameter of 47 mm, an ultimate strength (F_u_) of 1820 kN, and a yield strength (F_y_) of 1650 kN. A load cell was installed with the application of three embedded EM sensors to measure the accurate prestressed force. The load cell used in the experiment is a VW type (SJ-3000), made by Sungjin Geotec in South Korea. Its specifications are shown in Table 1. The VW type load cell uses a principle by which the vibration wire generates the resonant frequency, and the frequency is transmitted to the output device to display the necessary engineering unit when it is magnetized by the magnetic coil mounted due to the load. The embedded EM sensor and its specifications are shown in Figure 4 and Table 2, respectively. The bobbin was made of a primary coil part with a diameter of 117 mm and a secondary coil part with a diameter of 107 mm. The primary and the secondary coils were wound 300 times and 120 times, respectively.

Figure 5 shows the location of the temporary steel rods and the EM sensors. Forty-eight steel rods were installed in the pier at intervals of 500 mm from each other, and they were connected from the pier to the pier table. The EM sensors were installed on three steel rods, two of which were connected to the steel rods with the length of 10 m; the other one was connected to the steel rod with the length of 11 m.

Figure 6 shows the installation process of the EM sensors. It was decided that the locations of the EM sensors would be at three spots on the pier head, where two of them were eccentric sections, and the other was a midsection. The sensors were installed with an external sheath for the temporary steel rods. The cables connected to the sensor were pulled out of the bridge using a cable tube to prevent damage during the concrete pouring. After the concrete was hardened, a tensile force of 900 kN was introduced to all the temporary steel rods. The magnetic hysteresis curve was measured by installing a container for the measurement under the pier. The measurement was continuously conducted in 30 min intervals for approximately 10 months, from 17 March 2019 to 6 January 2020.

A module incorporating a voltage amplifier, a data acquisition device, and a desktop computer equipped with the NI (National Instrument) LabVIEW software package (Version 19.0) was used for the measurement. The measurements were performed five times to reduce the measurement error from each sensor.

### 3.2. Initial Value Calibration of Tensile Force

The initial value calibration results using the EM sensors are shown in Figure 7. The tensile force was measured six times by progressively increasing the prestressing force to 180 kN, 383 kN, 628 kN, 849 kN, 900 kN, and 915 kN. The area of the magnetic hysteresis curve was measured simultaneously with the prestressing process. 

The area of the B-H curve corresponding to the value of the tension force introduced to each steel rod could be obtained through the initial value calibration. The area of the curve increases as the tension force gets developed to the designed tensile force. Therefore, it was confirmed that the area of the B-H curve and the introduced tensile force are in positive correlation.

### 3.3. Measurement Results of the Field Experiment

The results of measuring the tensile force using the load cell are shown in Figure 8. The load cell data were used as the absolute values of the force introduced to the temporary steel rods. In addition, the results of the change in area of the magnetic hysteresis curve measured using the embedded EM sensors and the change in field temperature are shown in Figure 9.

During the measurement, the input voltage remained constant to a 0.02-Hz triangular wave of ±3 V. Although the voltage input value to the sensor remained constant, the area decreased as summer approached and then increased again as winter approached. The results of Figure 9 indicate that the area of the magnetic hysteresis curve measured by the embedded EM sensor exhibits an opposite trend to that of the temperature change.

### 3.4. Tensile Force Estimation with Temperature-Compensated Data

The temperature compensation method was described in the former section, and the technique was employed in the data-compensation process. The area of the measured hysteresis curve, which showed an inverse relation to the temperature data, was corrected through temperature compensation, and the results are shown in Figure 10.

The trend of the magnetic hysteresis curve due to the temperature change was modified from each sensor. It could be identified that the area decreased consistently in accordance with prestress loss during the measurement. This result agreed with the descending tendency of the tension force introduced to the temporary rods.

The tension force was estimated using the area of the magnetic hysteresis curve from each sensor after correcting by temperature compensation. The results of comparing the measured force with the load cell measurements are shown in Table 3. In addition, the sensor measurements and the load cell results are plotted in Figure 11.

## 4. Conclusions

The tensile force management of the temporary steel rods plays a key role at the phase of attaching segments during the construction of an FCM bridge to prevent any possible collapse and to meet the construction schedule. Therefore, a tensile force estimation method to measure the area of the magnetic hysteresis curve of the temporary steel installed in an FCM bridge was proposed in this study. For the measurement, the embedded EM sensors were installed in combination with a sheath tube outside the temporary steel rod. The magnetic field strength and the induced magnetic flux density were obtained to form the magnetic hysteresis curve. It was possible to estimate the tension by tracking the changes in the magnetic hysteresis of the steel rods. As a noticeable trend in the curve due to the temperature was observed, a temperature compensation technique using quadratic polynomial fitting was applied to offset the measurement error. Three EM sensors were employed to increase the reliability of the measurement data. To verify the applicability of the study, the experiment using the EM sensors was conducted at an actual FCM bridge construction site. Three EM sensors were installed at different spots on the pier head. An initial value compensation method was exploited as a reference by measuring the tensile force and the area of the magnetic hysteresis curve at the same time during the prestressing process. In the field experiment, steel rods with a diameter of 47 mm, an ultimate strength (F_u_) of 1820 kN, and a yield strength (F_y_) of 1650 kN were used; the bridge was a PSC box girder bridge with a total length of 640 m and a width of 24.51 m. The results of the tension estimation were compared to the tension measured using a load cell installed on a steel rod with sensor 1. The estimated results of sensors 1, 2, and 3 showed average error rates of 0.68%, 0.95%, and 0.82%, respectively. The field applicability of the long-term tensile force monitoring method using embedded EM sensors was verified through experiment and analysis. This study’s reliability would be further increased by acquiring more data through the long-term monitoring of additional bridge construction sites and by adopting more accurate methods to reduce the error rate. Based on this, it is expected to develop into a method for estimating the tensile force that can be applied to all PSC bridges to be constructed using FCM.

## Figures and Tables

**Figure 1 sensors-22-01005-f001:**
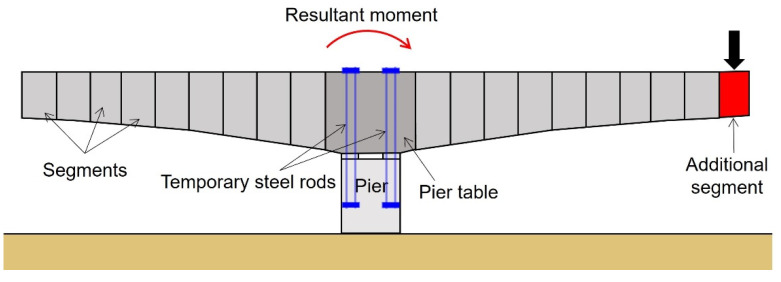
Illustration of unbalanced moment occurrence during construction of FCM bridges.

**Figure 2 sensors-22-01005-f002:**
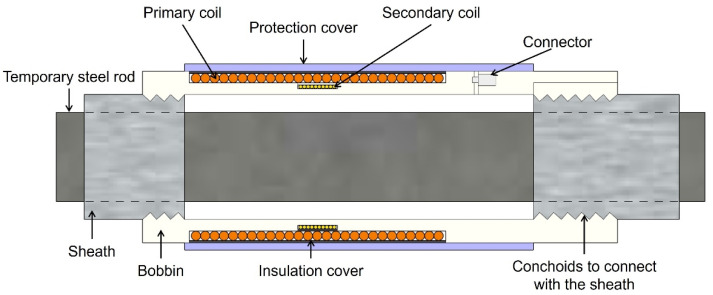
Schematic diagram of embedded EM sensors.

**Figure 3 sensors-22-01005-f003:**
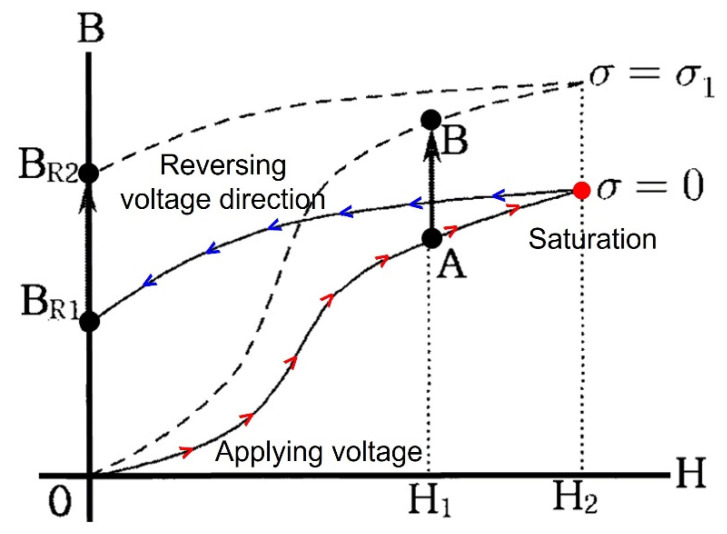
Change of magnetic hysteresis curve according to the effect of tensile force.

**Figure 4 sensors-22-01005-f004:**
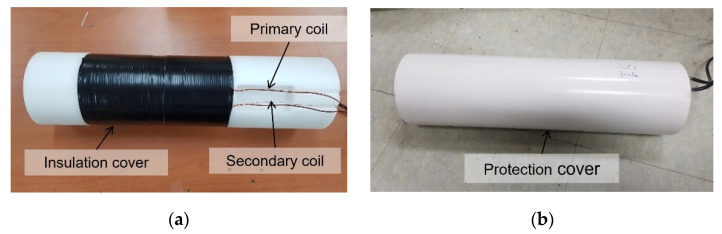
Fabricated EM sensor: (**a**) parts of EM sensor with insulation cover; (**b**) EM sensor equipped with protection cover.

**Figure 5 sensors-22-01005-f005:**
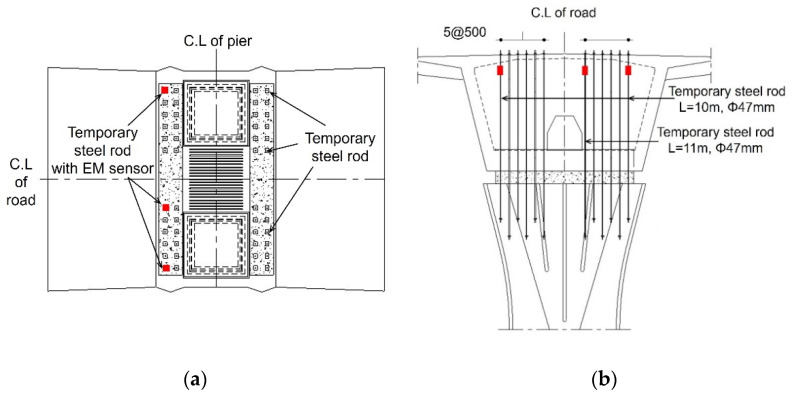
Layout of pier and pier table: (**a**) locations of EM sensors installed on pier table. (**b**) cross-section of pier and pier table.

**Figure 6 sensors-22-01005-f006:**
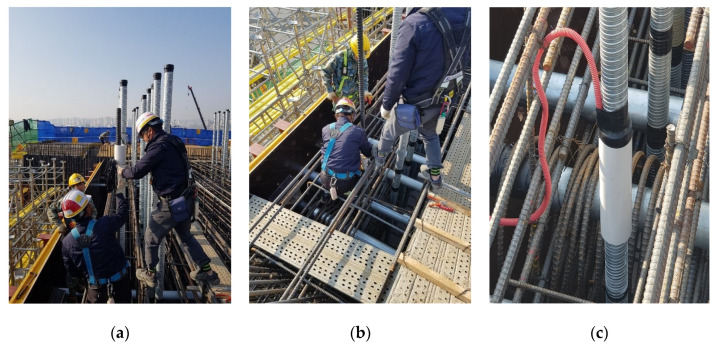
Installation process of embedded EM sensors: (**a**) inserting the sensor after cutting the sheath; (**b**) sheath tube and sensor combination; (**c**) electric wire protection with cable tube.

**Figure 7 sensors-22-01005-f007:**
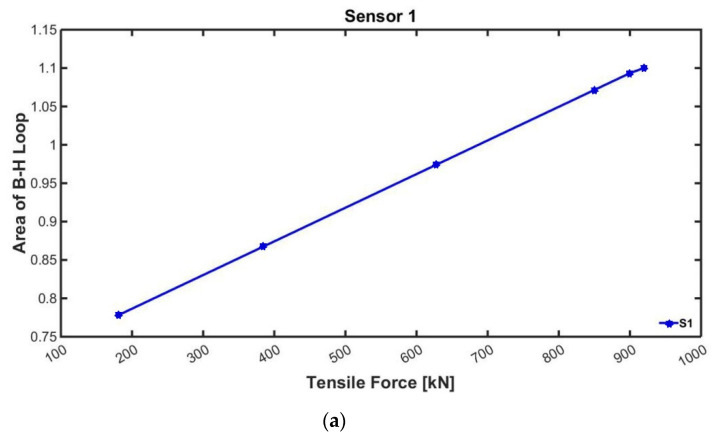
Calibration results using EM sensors: (**a**) Sensor 1; (**b**) Sensor 2; (**c**) Sensor 3.

**Figure 8 sensors-22-01005-f008:**
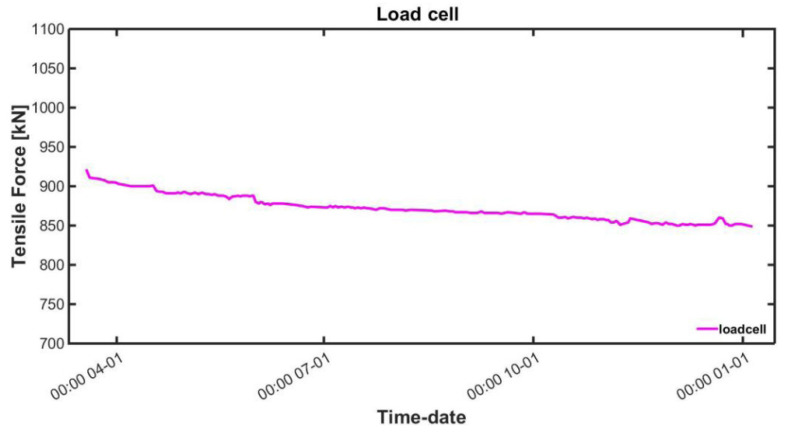
Tensile force measurement result using a load cell.

**Figure 9 sensors-22-01005-f009:**
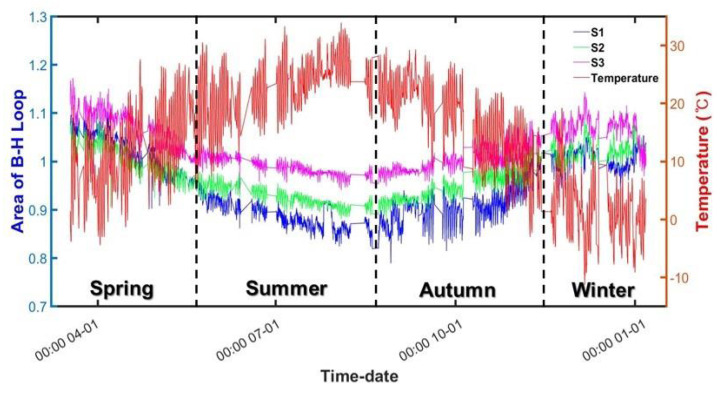
Changes in the area of magnetic hysteresis curves and field temperature during the measurement.

**Figure 10 sensors-22-01005-f010:**
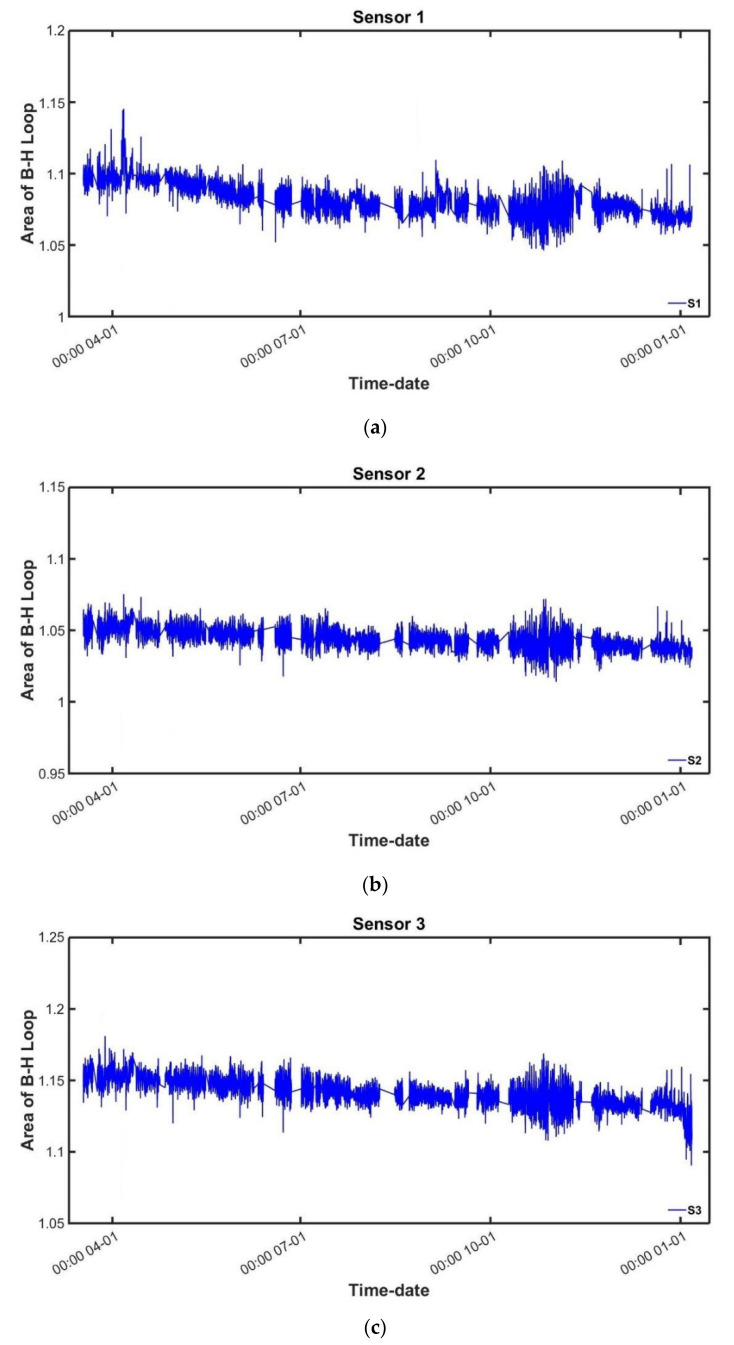
B–H loop area changes of sensors after temperature compensation: (**a**) Sensor 1; (**b**) Sensor 2; (**c**) Sensor 3.

**Figure 11 sensors-22-01005-f011:**
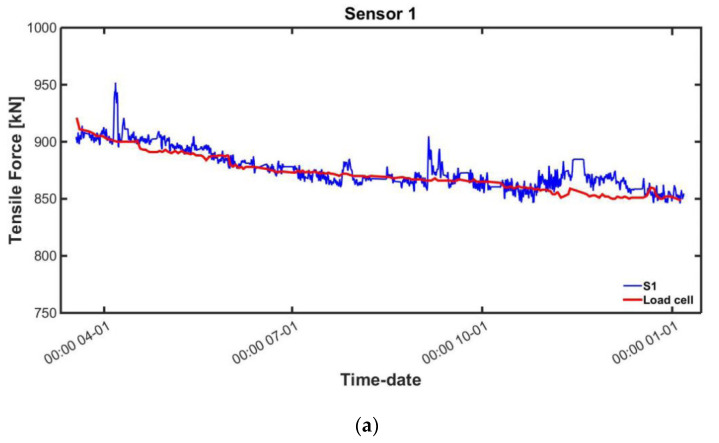
Estimated tension after temperature compensation and actual tension: (**a**) Sensor 1; (**b**) Sensor 2; (**c**) Sensor 3.

**Table 1 sensors-22-01005-t001:** Specifications of load cell (SJ-3000).

Classification		Values and Description
Capacity		1177 kN
Ultimate overload		150% of Capacity
Resolution		0.025% F.S.
Accuracy		±0.1~±1% F.S.
Linearity error		±0.5% F.S.
Material		SCM alloy steel
Gauge		3 VW Strain gauge (4 Strain gauge)
Thermal expansion coefficient		10.8 × 10^−6^/°C
Operating temp. range		−40 °C~80 °C
Temp. sensor	Type	NTC Thermistor (3KD-ATF)
	operating range	−40 °C~80 °C
	Accuracy	Thermistor: ±1 °C
Waterproof		Fluoride O-ring, High-density vacuum grease coating
Weight		4.95 kg

**Table 2 sensors-22-01005-t002:** Specifications of the embedded EM sensor.

Classification	Primary Coil	Secondary Coil
Diameter of bobbin (mm)	117	107
Diameter of coil (mm)	1.2	0.3
Number of turns	300	120

**Table 3 sensors-22-01005-t003:** Comparison of the estimated tensile force obtained using embedded EM sensors and the tensile force measured using a load cell.

Date		Load Cell (kN)	Sensor 1		Sensor 2		Sensor 3		Temperature (°C)
			Estimated Tension (kN)	Error Rate (%)	Estimated Tension (kN)	Error Rate (%)	Estimated Tension (kN)	Error Rate (%)	
1 April 2019	7:00	891	893.89	0.32	896.06	0.57	890.64	0.04	1.1
12 April 2019	14:00	880	884.04	0.46	884.10	0.47	880.23	0.03	13.0
13 May 2019	10:00	875	874.15	0.10	882.55	0.86	880.84	0.67	23.1
30 May 2019	9:00	872	866.71	0.61	890.79	2.15	881.98	1.14	18.6
14 June 2019	8:00	869	861.24	0.89	871.08	0.24	872.91	0.45	22.8
17 July 2019	10:00	865	860.11	0.57	864.91	0.01	871.28	0.73	25.5
14 August 2019	11:00	861	875.60	1.70	870.31	1.08	864.30	0.38	30.0
6 September 2019	9:00	858	858.15	0.02	863.12	0.60	864.65	0.78	23.2
23 October 2019	15:00	850	877.66	3.25	877.31	3.21	867.11	2.01	18.5
6 December 2019	14:00	852	847.68	0.51	844.30	0.90	828.37	2.77	4.3

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
