# Peer review of "Verification of Tensile Force Estimation Method for Temporary Steel Rods of FCM Bridges Based on Area of Magnetic Hysteresis Curve Using Embedded Elasto-Magnetic Sensor"

_sensors, 2022, doi:10.3390/s22031005_

Round 1

Reviewer 1 Report

The paper presents an embedded elasto-magnetic sensor which is used to estimate the tension of temporary steel bars in bridge construction, the effectiveness of this method is verified by the experimental data of construction site. The following concerns should be well addressed before a further consideration about the publication, and the revised one should be returned to the reviewer for evaluation again.

  1. The paper seems just presents the application of the method steel bar tension detection Lack of discussion of theoretical methods. What is the novelty? The author should clarify the contribution?
  2. The introduction is too wordy, The free cantilever method of prestressed concrete bridge is only a research background, It is proposed that temporary steel bar tension monitoring is an engineering requirement in bridge construction, There is no need to elaborate. The introduction should focus on the research progress of steel bar tension monitoring in recent years, summarize and compare the existing methods, find the engineering needs and limitations of existing methods, and lead to the method of this paper.
  3. This research subject is in the scope of structural health monitoring, however, the literature review of some excellent research work about bridge SHM is missing which should be added and referred in the manuscript, such as:

“Correlation-based estimation method for cable-stayed bridge girder deflection variability under thermal action, Journal of Performance of Constructed Facilities. 2018.”

“Design and application of structural health monitoring system in long-span cable-membrane structure. Earthquake Engineering and Engineering Vibration. 2019, 18(2): 461-474.”

  1. For the measurement error caused by temperature variation, the technical method of temperature compensation is put forward, but the second section lacks theoretical support.
  2. For the quadratic polynomial fitting for temperature compensation method to predict tensile force (formula one), There is no derivation and numerical analysis of his origin. Why do polynomials choose quadratic? In addition, how to determine the values of the second-order fitting coefficients in the equation in the actual application?
  3. There is only one type of embedded elasto-magnetic sensor used in the test. Does different types affect the accuracy of tension monitoring? In practice, how should the sensor type be selected when different steel rods have different tension values?
  4. Is there any criterion for the placement of the embedded elasto-magnetic sensors? In order to better reflect the advantages of temperature compensation, how to select the sensor positions?
  5. Sunlight has a great influence on the temperature of steel bar itself. How is the influence of light on temperature considered in this paper?  Is this problem considered when placing sensors?

9.This paper only compares the estimated tension with the tension measured by the load cell, and does not analyze the advantages and applicability of this method with the existing steel bar tension monitoring methods. In addition, from the data results, the estimated tension errors at different temperatures are different and irregular. Whether is this situation caused by the fitting of quadratic polynomial by temperature compensation method itself or other reasons?

Author Response

Comment 1

Answer: We found many studies about monitoring the tensile force PS tendon in bridges in service. Most techniques and methods were applied to the structures in service. We focused on the monitoring the tensile force of temporary steel rods that are going to be removed after completion of the construction. Moreover, temperature condition of steel rods and the sensor, which are sensitive to the change of temperature, was considered to investigate the effect of temperature on the output signal over a long period of construction.

Comment 2

Answer: Various studies to measure the tensile forces were described in the introduction section including FBG sensor, EM sensor, magnetic flux sensor, and so on. An additional description about why the tensile force monitoring is necessary in the construction stage of FCM bridge was added to enhance our research background.

Comment 3

Answer: Two recommended papers were good enough to be referred. Excellent results were shown in those papers about SHM of bridge structures. We described methods in those papers in the introduction section to compare with.

Comment 4

Answer: The change in area of the graph of the measured magnetic hysteresis curve using embedded EM sensors was approximated to the second-order through regression analysis considering the correlation between the temperature and the output voltage. The temperature data measured in the field was substituted into the equation with respect to the measurement time.

Comment 5

Answer: Temperature of sensors could be measured and the output voltage was obtained simultaneously corresponding to a specific value of tensile force. The correlation between the temperature and the output voltage was not linear, which led us to choose the quadratic polynomial fitting.

Comment 6

Answer: As the designed tensile force for all temporary steel rods was same as 900 kN, one type of EM sensor was used for the magnetic hysteresis measurement.

Comment 7

Answer: As we expected the temperature might affects the output signal of EM sensors at first, two of the sensors were located at both sides of the pier table, and the other was located at the middle of the pier table. However, the results showed that there were not significant effects on the data according to the sensor positions.

Comment 8

Answer: Sunlight might affect greatly the steel rods increasing the steel temperature. To prevent that, we put the wooden box over the exposed part of the steel rods to minimize the effect of sunlight on the steel rods.

Comment 9

Answer: We used the atmospheric temperature at the site. However, we could’ve get better results if we had measured the temperature of the temporary teel rods. Nevertheless, it can be said that reliability has been proven to be good because the maximum error occurred within ±4% compared to the tension measurement using the load cell, and the average error rate was within 1%. In the future, we will consider conducting research by measuring the temperature of the target specimen itself.

Reviewer 2 Report

The Authors have investigated thoroughly a very pertinent and interesting practical problem faced in civil constructions. Application of Elasto-Magnetic sensor was made to monitor tensile force in FCM steel rods for duration of about 10 months. A clear relation between area under B-H curve and applied load was demonstrated.  Overall, I recommend an acceptance once the followings are implemented/:

  1. Details of EM sensor and load cell like make, model and capacity are missing. It is unclear whether the Authors used a commercial EM sensor or a home-made one.
  2. In figure 2, yellow colour arrows may be changed to dark colour as the contrast against white background is poor. In addition, different colours can be used for better illustrate primary and secondary coils.
  3. A single schematic diagram of the three sensors’ locations and inclinations around steel rod can be included.

Author Response

Comment 1

Answer: The EM sensor was manufactured by researchers in this study, and a commercial product was used for the load cell. The contents showing the structure and principle of the EM sensor were added to 2.1 EM sensor, and the prefabricated EM sensor is shown in Figure 4. In the case of the load cell, the capacity and detailed specifications of the products used are shown in Table 1.

Comment 2

Answer: The original figure 2, we changed the color of the indicating arrow to black for contrast against the white background. In addition, different colors were used to draw the two types of coils. It was shown in the Figure 2.

Comment 3

Answer: As commented, a single schematic diagram indicating the locations of three sensors and steel rods were included in Figure 5 referring to the original construction drawings.

Reviewer 3 Report

The manuscript entitled “Verification of Tensile Force Estimation Method for Temporary Steel Rods of FCM Bridges based on Area of Magnetic Hysteresis Curve using Embedded Elasto-Magnetic Sensor” proposed a tensile force estimation method for temporary steel rods of the free cantilever method bridge using embedded Elasto-Magnetic sensors. This manuscript can be accepted as a technical report, not an article. Otherwise. The authors mustprovide more and detailed discussions for the recorded results. Technical Comments:
1- The technical writing and grammar are not good and need more improvement.
2- Line 27: It should be "in", not "on".
3- Line 36: Is this above or below error? I mean is it +4% or -4%?
4- Lines 62-63: This reviewer suggests providing an illustration of this technique in construction and what is the position of this steel rod relative to the pier. Also, showing the developed internal forces (moments) will be great.
5- Line 139: The term "V_T" should be corrected to match that one in equation 1. Also, the term "T" should be defined. Is it the change in temperature or time?
6- Line 147: This sub-section should be moved to Section 2.
7- Lines 150-152: Layout of the bridge including the location of the steel rod should be provided.
8- Line 153: What was the length of these steel rods?
9- Figure 2: This figure is not clear and must be revised. Where are the sensors and steel rods? The dimensions on the right side of the figure should be modified. I highly recommend providing an on-site photo of the embedded sensors before adding the protection.
10- Section 3.2: The main purpose of this section should be highlighted. This reviewer understands that the calibration step is to validate or verify the reading results.
11- Line 187: No need to add "respectively" at the end of this sentence.
12- Figure 4: The reading points should be shown in the Figure.

Author Response

Comment 1

Answer: We improved technical writing and grammar by including additional explanation about the sensor used in this paper and included the supporting information about the experimental setup at the site.

Comment 2

Answer: Corrected

Comment 3

Answer: Corrected to ±4%

Comment 4

Answer: The process of developing of the internal moments during FCM bridge construction was described in the Figure 1 (added).

Comment 5

Answer: V_T was corrected to VT, and the term ‘T’ was defined to ‘ on-site temperature’ .

Comment 6

Answer: Chapter 3 is about experimental study. So, We added a part for ‘2.1 EM sensor’ and made a schematic diagram of the EM sensor.

Comment 7

Answer: The layout of the steel rods were included in Figure 5(newly added).

Comment 8

Answer: The length of the steel rods was included in above the Figure 5( Layout of pier and pier table)

Comment 9

Answer: The locations of the EM sensors and steel rods were presented in Figure 5 (layout of pier and pier table). The photo of fabricated EM sensors which was used in this study was presented in Figure 4(newly added)

Comment 10

Answer: Supporting explanation about ‘initial value calibration of tensile force’ was added in the section 3.2. It presents the relation between the hysteresis curve and the measured tensile force by a load cell.

Comment 11

Answer: Corrected

Comment 12

Answer: The reading points were presented on the graph of Figure 7.

Round 2

Reviewer 3 Report

The authors have addressed most of the reviewer's comments and the manuscript can be accepted for publication.